# “What You Leave…Will Leave You”: A Qualitative Study of Perceptions of Midwifery’s Intangible Heritage and Professional Identity Among Midwives and Student Midwives in Cyprus

**DOI:** 10.3390/healthcare13151936

**Published:** 2025-08-07

**Authors:** Maria Panagiotou, Eleni Hadjigeorgiou, Stavros Vryonides, Maria Karanikola, Anastasios Merkouris, Nicos Middleton

**Affiliations:** 1Department of Nursing, School of Health Sciences, Cyprus University of Technology, Limassol 3036, Cyprus; eleni.hadjigeorgiou@cut.ac.cy (E.H.); maria.karanikola@cut.ac.cy (M.K.); anastasios.merkouris@cut.ac.cy (A.M.); 2School of Life and Health Sciences, University of Nicosia, 46 Makedonitissas Avenue, Nicosia 2417, Cyprus; vryonides.s@unic.ac.cy

**Keywords:** experiences, midwifery intangible heritage, midwifery profession, perceptions, professional identity

## Abstract

Background: Midwifery’s Intangible Heritage was officially recognized by the United Nations Educational, Scientific and Cultural Organization on 6 December 2023, highlighting that elements of midwifery knowledge and practice, shaped over generations, constitute cultural heritage worth safeguarding. While previous studies have investigated midwives’ perceptions of professional identity, none have done so within the explicit framework of MIH. Objective: this study explored how midwives and student midwives in Cyprus perceive the intangible heritage of their profession and how it relates to their shared professional identity. Methods: A qualitative descriptive study was conducted between April and July 2023. Three focus groups were held, involving 22 participants: 15 registered midwives and 7 student midwives. A semi-structured interview guide consisting of 10 questions was used, developed by the lead author (M.P.) based on the literature and improvisation and finalized with the research team (E.H., S.V., N.M.) after expert input. Thematic analysis was performed inductively to identify recurrent themes. Results: Four major themes emerged: (1) key elements of Midwifery’s Intangible Heritage and their transmission across generations; (2) a sense of shared professional identity; (3) perceived threats to the midwifery profession; and (4) midwives’ expectations for the future of the profession. Conclusions: The findings reflect the historical background of midwifery in Cyprus and its contrast with contemporary practice, particularly within the context of the overmedicalization of birth and societal perceptions of midwifery in the socio-cultural setting. Safeguarding Midwifery’s Intangible Heritage requires both empowering women to seek midwifery-led care and enabling midwives to practice autonomously within their full professional scope. In addition, policymakers and educational bodies must support the preservation of midwives’ core skills through targeted educational curricula, structured mentorship, and continuous professional development.

## 1. Introduction

The authentic form of knowledge, unique to each region, has historically emerged from the need for survival. These knowledge systems are dynamic and constantly changing and include the experiences, beliefs, values, and practices that a local community has developed over time, known *as attitudes*, *knowledge*, *skills*, *and techniques*, which are recognized as a developmental factor of these local communities [1]. Within this framework, midwifery practice and knowledge were acknowledged as part of Midwifery’s Intangible Heritage (MIH) by the United Nations Educational, Scientific and Cultural Organization (UNESCO) on 6 December 2023 [2]. Common professional identity is defined as the conceptual state that describes how we perceive ourselves within our professional context and how we communicate this to others [3]. While a number of studies have explored the perceptions of midwives about their common professional identity, none of them have done so specifically in the context of MIH.

In Cyprus, midwives have been historically recognized as skilled women with adequate knowledge to care for pregnant and birthing women [4]. In the absence of a formal health care system, midwives were the sole “representatives” of an informal care system providing healthcare services to the community, witch today would be considered Primary Health Care [5]. Formal midwifery education in Cyprus began in 1932, with the first trained midwives being registered under the initial Midwifery Law and listed in the country’s first registrar body of health professionals. In 1951, a structured educational program was introduced, allowing students to study midwifery after completing high school, through the newly established Nursing and Midwifery School. A major shift occurred in 1980, when direct entry into midwifery was discontinued. From that point onward, midwifery education became accessible only to those who had first completed nursing studies. Since 2016, midwifery education in Cyprus has been offered exclusively at the postgraduate level through a two-year Master’s Degree in Midwifery, following a four-year Bachelor’s in Nursing. Compared to the past, midwifery today has undergone significant professional and academic development, and Midwives in Cyprus are now scientifically qualified and well-trained health professionals [4]. Midwifery in Cyprus is governed by legislation (Nursing and Midwifery Law 1988/2020basic and amending) that is fully aligned with European directives and recognizes midwives as autonomous professionals responsible for the care of women monitoring, support, and counselling throughout pregnancy, conducting normal births on their own responsibility, and providing care to both mother and newborn during the postpartum period [6,7]. This framework reflects key international policies, including the WHO’s Strategic Directions for Nursing and Midwifery (2021–2025) and the ICM’s Essential Competencies, both of which support midwifery-led models of care and the midwife’s role as a primary provider for low-risk pregnancies [8,9]. Together, these policies provide a strong foundation for strengthening midwifery care in Cyprus and ensuring that midwives are empowered to contribute to safe, equitable, and woman-centered maternity services.

However, the increasing rates of caesarean sections (C/S) in Cyprus, from 56,9% in 2016 and 55,8% in 2020 to 62% in 2022 and 60,7% in 2023 [10,11], have led to a decline in normal births, and a shift in the role of midwives. This increasing trajectory over the last years raises concerns with regard to the overuse of medical interventions in low-risk pregnancies, which often deviates from international recommendations. Currently, there is no official classification system for C/S (e.g., Robson). In official perinatal statistics, around 30% of C/S are recorded as “emergency” while the remaining 70% are recorded as “elective” without further breakdown. This has led to the development of a National Strategy for the reduction of C/S by the Ministry of Health, but it is too soon to see any results from its implementation.

Consequently, midwives often face significant challenges when advocating for normal childbirth, such as the prevailing emphasis on medicalized approaches, limited opportunities to lead physiological births, and pressure to adhere to obstetric-led decisions. Additionally, the dominance of the medical model frequently leads to the marginalization of midwives’ expertise, particularly in private sector settings. These challenges are further compounded by inadequate institutional support, including understaffing, the absence of midwives in leadership roles, and minimal involvement in clinical policy-making. The findings of a study from Cyprus over a decade ago read like a warning that remains deeply relevant today: “In Cyprus, midwifery in dying” against a continuing climate of ‘physician dominance’ [12].

Following the recent healthcare reform in Cyprus, access to community midwifery has been formally recognized as a right for women to seek midwifery care and a responsibility of the system. Under the General Healthcare System (GeSY), up to six midwifery visits are now covered, marking a significant step toward integrating midwives as autonomous healthcare professionals, in line with existing legislation [13]. While not covering the full scope of midwifery practice, this development offers a renewed chance for women to reconnect with midwives as trusted health professionals, particularly through their vital educational role in antenatal care [14]. However, in practice, access to such care—especially in the private sector—is often left to the patient’s discretion, and many women are unaware of their right to receive postnatal follow-up from a midwife.

This study aimed to explore the perceptions of midwives and student midwives in Cyprus about the intangible heritage of the profession, as well as the sense of their professional identity that connects generations of midwives.

## 2. Materials and Methods

A qualitative descriptive study was conducted between April and July 2023, using a series of focus groups to explore midwives’ perceptions of MIH and sense of common professional identity. A focus group methodology was chosen to capitalize on group dynamics and participant interaction during the group discussions, allowing insights to emerge that may not have surfaced through individual interviews or other forms of information gathering [15].

### 2.1. Sampling

Purposive sampling was used to recruit participants who could offer diverse insights focusing on generational perspectives and experiences across care settings. Two focus groups, 6 and 9, registered midwives, respectively, as well as one with 7 student midwives, were conducted. This is in line with recommended group sizes of 6–10 participants for optimal depth and interaction [16]. Research shows that approximately 80% of key themes emerge within two to three focus groups; therefore, our three focus groups fall within these empirically supported ranges, resulting in a methodologically justified and adequate sample size for qualitative thematic analysis [17]. Saturation was assessed during the analytical process, with no new themes emerging by the third focus group.

While student midwives do not yet bring professional experience to the discussion, a separate focus group was conducted with them to capture their distinct educational perspective and evolving professional identity. Note that this was the entire student cohort at the time, reflecting both the small size of the country and the structural challenges of training midwives in a health system characterized by high caesarean section rates. Keeping the student group separate allowed for a more open and relevant discussion among peers, free from the influence of hierarchical dynamics that might arise in a mixed group, while still contributing valuable insight into the continuity and future of the profession.

For the registered midwives, purposive sampling was employed to ensure the inclusion of a mix of participants with diverse characteristics in terms of years of experience and practice settings, thereby providing the opportunity for varied experiences and the necessary heterogeneity to generate rich and meaningful data [18]. Participants were contacted either through personal communication or via social media platforms, specifically Messenger and Viber, using professional midwifery networks and peer group chats. Focus group size was determined with the aim to facilitate adequate attendance and active participation by all members, however, taking into account participants’ availability and preference, distance needed to travel to the site and working shifts [19], the SM focus group was conducted in an academic classroom and the Midwives focus groups were conducted in Limassol’s General Hospital amphitheatre.

The inclusion criteria for the study were (a) registered midwives who were willing to participate and were working in public or private maternity units, in midwifery administration, or within the midwifery services of the Ministry of Health; and (b) student midwives enrolled in the midwifery program at the Cyprus University of Technology, which, at the time of data collection, was the only institution actively offering midwifery education in Cyprus. The exclusion criteria were (a) midwives or student midwives unwilling to participate; and (b) midwives not registered in the official Registry of Midwives, not currently practicing in maternity-related roles, or students not enrolled in midwifery studies during the data collection period.

### 2.2. Topic Guide

The semi-structured interview guide was initially developed by the lead researcher (M.P.) following a literature review and a reflective process based on professional experience, initially mapping the main challenges that the midwifery profession faces [20]. The draft version was then reviewed and refined in collaboration with the research team (E.H., M.Pa., N.M., and S.V.) to ensure that the questions were relevant, clearly worded, non-leading, and participant-oriented. Further modifications were made following internal discussion, resulting in a finalized guide consisting of 10 questions grouped into three thematic areas [20,21]. The research team was also asked to reflect on any additional questions necessary to cover the key themes. Following discussions, further modifications were made, including the addition of a brief introductory section to provide a smooth introduction of the topic to the participants. The agreed final version of the guide, consisting of 10 questions grouped into three thematic areas, is presented in Table 1.

### 2.3. Data Collection

The three group discussions were conducted at different locations as necessary, such as the workplace conference room and the university classroom, prioritizing convenience for the participants while ensuring an atmosphere of comfort, privacy, and trust [18]. Each session was audio-recorded with participants’ written consent and lasted between 85 and 95 min, producing 109 pages of transcribed material. The discussion in all focus groups flowed naturally, and the participants appeared comfortable expressing their views openly. If some opinions differed, the participants listened respectfully to each other’s different points of view.

Although the number of focus groups was pre-determined based on resources and practical considerations, data saturation appeared to be reached by the third focus group, as no new significant themes emerged. However, despite the inclusion of participants with varied professional experience, it remains unclear whether theoretical saturation was also achieved, given the purposive sampling approach.

Several steps were taken in order to ensure the methodological rigor of the study from the data collection process. Two experienced moderators conducted the sessions: E.H., an academic midwife, facilitated the group with student midwives, while S.V., a registered nurse, facilitated the groups with practicing midwives. Both moderators had a lot of prior experience in focus group facilitation and were impartial with no vested interest in the participants’ responses, so the participants felt free to express their opinion and engage in the discussion from the early stages of the interview [15]. An observer (MP) was also present during the interviews, who kept notes of non-verbal responses and interactions. In addition, the observer noted the level of involvement among the group and the style of the moderators in order not to exert high influence or control on participants’ responses during the process of the interview [15]. At the end of each focus group, the facilitation team conducted a debriefing session to reflect on the process and discussion.

### 2.4. Data Analysis

The data were analyzed using a thematic analysis approach in order to achieve a representation of the whole in a detailed and systematic manner [13], involving a series of steps from the transcription of the discussions to data coding and synthesizing categories into themes. Briefly, interviews were manually transcribed verbatim and checked by the research team in order to produce an accurate record of the discussion [22]. They were analyzed together since the aim was not to draw inferences about differences between student and registered midwives and was performed in parallel by two researchers (MP/SV and MP/EH), in a systematic iterative process, following a set protocol, with the involvement of a third researcher (NM) where necessary and giving the opportunity for reflection and discussion among whole research team [19]. Each transcript was read several times to familiarize with the content and was hand-coded in parallel and independently by two researchers, with the involvement of a third researcher (NM) where necessary, giving the opportunity for reflection and discussion among the whole research team [19]. To ensure process transparency, researchers compared their coding, discussed and debated their coding regularly, and identified any discrepancies and differences until consensus was reached. Steps and actions are presented in Table 2.

Following the initial coding, similar codes with the corresponding quotes were grouped into 15 preliminary categories. Unassigned codes were revisited to assess whether the categories could be adjusted to provide a more comprehensive and accurate representation of the codes or excluded from the analysis. The aim of the iterative analytical process was to synthesize the categories into themes while maintaining a direct link to the raw interview data, progressing from a primarily descriptive level to a higher-order latent level of interpretation. This process led to the development of four themes.

## 3. Results

The sample included midwives with varying years of professional experience, ensuring representation from both clinical practice and midwifery management and leadership. Student midwives were recruited from the Cyprus University of Technology, one of only two institutions offering midwifery programs in Cyprus.

A total of 22 participants took part in the focus groups: 15 registered midwives and 7 midwifery students. Participants were distributed across three focus groups: group A: *n* = 7 midwifery students; group B: *n* = 6 midwives; group C: *n* = 9 midwives. Twenty-one of the participants were female and one was male.

The students’ group was homogenous in terms of education and midwifery practice level. All midwifery students participating were Registered Nurses in their second year of a midwifery postgraduate qualification. The two midwives’ groups were heterogeneous in terms of both professional experience and academic qualifications. Among Registered Midwives, age ranged from 34 to 65 years (45.5 ± 11.25), and professional experience in midwifery ranged from 6 to 40 years (23 ± 6.5). Values in parentheses indicate mean ± standard deviation (Table 3).

Using an inductive thematic analysis approach, 15 codes were generated and organized into four main themes, each supported with illustrative quotes. A Thematic Analytic Schema is also provided to draw the links between categories and themes, as presented in Figure 1.

### 3.1. Themes

#### 3.1.1. The Elements That Constitute MIH and Their Transfer Through Generations

Participants had no difficulty articulating their understanding of what MIH means. There was agreement that such intangible heritage exists and it is transferred from one generation of midwives to the next, as part of their shared history and culture. They emphasized that birth is a natural, physiological event, and supporting it is the very purpose of the existence of midwives. One midwife articulated this clearly:


*“… perpetuation of the species is a stable event in human life, and midwives are always needed” … “Society is changing but the MIH, such as being close to the woman, caressing her and talking to her does not change and no one can take it from the Midwife”.*
(M1)

Participants described numerous subtle and embodied practices—skills, patience, touch, eye contact, calmness, serenity, confidence, physical presence, and empowerment of women—that are adopted and passed on through professional socialization. One of the most important elements of MIH is the transmission of practices, experiences, and techniques from “old” midwives to “new incomers” to the profession. 

As one participant explained:


*Unless you see the old ones apply them and learn, you cannot have the adequate experience.*
(M6)

Furthermore, the importance of experienced midwives serving as role models for the younger ones was emphasized:


*Oldest midwives should embrace the young ones. To be mentors.*
(M7)

However, one midwife noted that, nowadays, the younger midwives often lack the opportunity to learn directly from senior midwives, instead relying heavily on physicians’ instructions: 


*[they] follow physicians’ orders and practice because they DON’T have experienced midwives to follow.*
(M4)

Student midwives echoed that experience is fundamental in practicing midwifery, describing it as essential for developing professional confidence, gaining respect, and autonomy in decision-making. As one of the student midwives shared with the group: “Experience gives you strength and confidence” (SM2), emphasizing the need for mentorship from experienced midwives. Another one explained, “experience brings confidence in technique and continuous development is based on old standards” (SM3).

For student midwives, learning how to build a trusting relationship with the women they care for is not something they can be taught in the classroom, but only learn in practice, by observing and working alongside other midwives.

*A midwife is a person of trust … to be trusted by women who asked for her during labour. I could not understand exactly how to create this trust, until I started my practice and then I felt that indeed during the time of delivery a woman is together with a midwife*. (SM 4)

However, participants recognized that autonomy—a defining characteristic of traditional midwifery practice— is becoming increasingly difficult to preserve and transmit, largely due to the dominance of a highly medicalized birth environment. Despite these challenges, the participants viewed the preservation and transmission of professional independence as vital to the profession and expressed a sense of duty to safeguard and strengthen it further:


*I believe that the most important thing is our independence. In other words, I learned from the old midwives how to be independent and empower myself. And when we, the youngest ones, also try… Yes, we want to take it further… And we will take it….*
(M 6)

#### 3.1.2. The Sense of Common Professional Identity

The majority of participants expressed positive feelings about being a midwife, agreeing that there is a sense of common professional identity among them. They primarily attributed this shared identity to “a common cause” and “a shared responsibility,” centered on prioritizing the well-being of mothers and newborns:


*We are thinking about the good of the mother and child and we put everything … aside … woman and child are coming always first.*
(SM 1)

A fundamental element of midwives’ sense of common professional identity is the commitment to the physiology of birth as a “unifying” principle that transcends generations and geographical boundaries. This was repeatedly emphasized by participants in the discussion as a defining characteristic of midwifery practice at the core of the professional identity:


*What unites us all midwives like this, is that if you travel and go anywhere and find midwives, what unites us immediately… is the question if they have a lot of caesareans? What unites us is the concept that we all midwives believe in normal birth, and we are disturbed by the consequences of medicalization and deviations from normal birth and we fight to bring and keep birth in its normal state, as the old ones did… They wanted to give to the woman the choice to give birth naturally and we must keep it in this way. It characterizes midwives… it’s characteristic…*
(M1)

All of the participants expressed deep pride in their identity as midwives (“very proud of being midwives”). They associated the title “midwife” with feeling “confident and proud”, always “introducing themselves as midwives”, even in a personal context.

Beyond pride, participants saw midwifery as distinct from other health professions, defined by the ability to act independently under pressure.


*Midwives are different from other health professionals because they have to think, decide and act quickly. They can’t be late. You evaluate and decide immediately.*
(M2)

In fact, in their descriptions, participants described midwifery not only as being associated with the act of natural birth but also emphasized their role and commitment in actively safeguarding the natural birth process. As the use of the term “saving” in the following quote suggests, this may often be a daily struggle, as a result of working in a medicalized birth environment, which is something that gives them a great sense of pride.


*All midwives felt pride in saving a woman from an unnecessary caesarean section or an intervention.*
(M7)

Women’s positive experiences provide midwives with a sense of purpose, which is reinforced over time through the recognition and gratitude they receive from women they have supported—often years after the birth. This appreciation brings emotional reward and strengthens their professional identity.


*There was a lady with her 20 years old son…she saw me and run towards me and hugged me…‘You delivered my son….i can never forget you’. Her son kneeled and kissed my hands. He said…‘these hands are the ones that touched me first…I honor them’…*
(Μ2)

This positive feedback not only affirms midwives’ role but also reinforces their ability to empower other women, with the emotional connection and trust during birth becoming a source of mutual strength.


*She said to me…I will never forget your eyes, the way you stared at me and said… you will make it. And I believed you, and I made it. I will never forget that.*
(Μ2)

Furthermore, experienced midwives play a critical role in empowering the younger colleagues and student midwives who seek opportunities to lead the births. To cultivate that sense of autonomy through the training process, they need to show trust in them while giving the right advice at the right time. This sense of shared strength—described as the “power of the maternity ward” by one of the participants—was fostered through trust and responsibility.


*In the past, as students, we were left alone to do so many things, and experienced midwives were there if we needed them. By this way, we felt so strong. This strengthened us. We felt part of the team, the power of the maternity ward.*
(Μ14)

#### 3.1.3. Perceived Risks That the Midwifery Profession Is Facing

The participants expressed skepticism regarding the ability to pass on the legacy of MIH to future generations. They highlighted the increasing medicalization of pregnancy and birth, along with an overreliance on technology, as major obstacles.


*I can remember that in older times a midwife was autonomous, strong, and independent and…alone. I was practicing alone at my village, and other midwives too… now, our issue is medicalization.*
(M3)

A key concern was the sharp decline of normal births and the persistently high rate of caesarean sections in recent years in Cyprus; for many of the participants, the current situation was a crucial turning point with serious consequences not only for the actual ability to train future midwives in numbers but also for the retention of essential skills among both midwives and obstetricians.


*Reduction in normal births leads to a reduction of skills. So, this thing creates insecurities, creates a reduction in training for new midwives, a reduction in skills, a reduction in everything that is a vicious circle that will not stop.*
(M9)

In combination with the growing emphasis on the academic dimension of Midwifery education, these phenomena have further reduced opportunities for gaining experience through clinical practice. This shift risks eroding not only the practical skills but also the deeper culture of midwifery that can only be transmitted through close interaction with experienced midwives in real-life settings. Characteristically, one of the participants warned about the risk of this irreversible loss:


*The young midwives took from the old knowledge and experience. The increase in knowledge at the university level has reduced the practical part that will be useful to you in times of need. **What you leave…will leave you**.*
(M9)

Another significant concern raised by the participants was that midwives, particularly those working in the private sector, often feel professionally limited and less visible in their role, frequently perceived and referred to as doctors’ assistants.


*During a night shift i was talking to a female patient, she was a teacher, about our role as midwives, that we do normal births …she was surprised…she thought we are just doctors’ assistants. And this is the majority’s opinion.*
(M9)

Another problematic area identified was the relationship with other health professionals, especially obstetricians. Participants perceptions were reported experiencing a lack of respect and even described a climate of fear around expressing independent clinical opinions, “Physicians’ don’t respect us and don’t listen to us as much as they should” (M10), particularly in the private sector, where midwives often feel that they cannot express opinions due to fear of professional repercussions.


*Because they know that they will be employed and paid by an obstetrician, and if they express a different opinion they will be fired.*
(M5)

This power imbalance impacts their autonomy and the ability to provide woman-centered care as they are “not able to build a relationship of trust with women” under these conditions.

Participants also expressed disappointment with the lack of support they received from the Nursing and Midwifery Hospital Administration and from the broader Organization Management. They described feeling both professionally underestimated and severely understaffed.


*They need to understand…first of all the administration of the hospital…If they cannot understand us…how do we expect to be understood by others.*
(M3)


*First of all, we are understaffed and we can’t do our job. When there are two people working, how am I going to be next to a birthing woman and support her? I don’ t think that they understand what we are doing … we feel left out, we are alone. I’m not sure what’s going on in the end, but we’ve been side-lined.*
(M5)

#### 3.1.4. Midwives’ Expectations About Their Profession

Participants discussed various strategies for strengthening and promoting the profession. A central theme was the need to actively reintroduce midwives to society and showcase the profession’s full scope:


*We need to reintroduce ourselves to society as midwives.*
(M10)


*We have to show off our profession and what we are able to do.*
(SM4)

Antenatal education was highlighted as a crucial investment for empowering women and promoting normal birth. Participants emphasized that education can change women’s perceptions and reduce fear:


*With antenatal classes we saw women change their perceptions. They started out telling me I will come [to the classes] but I am going to have a C/S, and after hearing about normal birth from us and from other women who had given birth, they became passionate about it. And that was the miracle of education.*
(M10)


*[we need to] empower women through antenatal education so that they can demand their rights for normal birth.*
(M7)

Participants emphasized the need for public awareness efforts that start early in life, proposing educational initiatives in schools to familiarize children with the natural birth process and the role of midwives, thus shaping the informed choices of future parents:


*We must go to schools and inform children about normal birth and breastfeeding, us midwives with experience in normal birth…what an investment for the future.*
(M1)

Another aspect discussed was the development of the profession. They expressed the belief that the recent establishment of community midwifery services within the current healthcare system holds the key to increasing women’s awareness of the role of midwives and their services, especially in providing postpartum care.


*We have to investment in post-partum care. It is an area of maternity care where the midwife’s role has not been fully integrated or emphasized in the current system.*
(M13)


*We have to go out to the community to regain our role and our autonomy back.*
(M10)

Improving midwives’ knowledge and practice through ongoing education was considered essential, both for self-development and also for the effective transmission of skills and evidence-based practices to newer generations.


*The experienced midwife should improve herself to pass on knowledge to and empower the new midwife. Be an example to follow.*
(M8)


*An experienced midwife with evidenced based knowledge and skills is fundamental for student midwives.*
(SM1)

To protect and strengthen the profession, participants emphasized the need for system change, starting with proper and full implementation of the Nursing and Midwifery Law and the European Directives, which explicitly allow midwives to care for women with low-risk pregnancies and their newborns.


*Full implementation of the Legislation is what we need.*
(SM3)

There was also a call for targeted support for midwives who work in the private sector and lack autonomy.


*Midwives working in the private sector need empowerment, reinforcement and support to get ahead.*
(M2)

Participants highlighted the critical role of women themselves in driving change:


*…empower women to assert their rights about normal birth and ask for a midwife to help them.*
(M1)

## 4. Discussion

This qualitative study offered insights into how midwives and student midwives perceive MIH, and how intergenerational knowledge, attitudes, skills, and practices contribute to shaping a shared professional identity. The findings suggest that medicalization, societal perceptions, and weakened leadership structures have significantly influenced the current status of the midwifery profession in Cyprus.

Participants emphasized the pivotal role of experienced midwives as mentors and role models. Recognized as experts in physiological birth, they play a central role in caring for women giving birth [23], and they also have a long history of mentoring students and new graduate midwives [24]. In parallel, midwifery students during their education identify role-model midwives and assimilate their attitudes and behaviors into a role model that represents their ‘ideal midwife’, to learn from her skills, attitudes, behaviors, and trust in physiological childbirth [25]. This close connection is necessary for learning [26], developing autonomy, confidence and competence in their clinical practice [27,28,29], and learn “how to be a midwife”, cultivating a sense of professional identity [30]. Midwifery education should integrate structured mentorship programs that connect students with experienced midwives, enabling the transfer of practical skills, professional values, and cultural competence. Embedding elements of midwifery’s intangible heritage within formal curricula can strengthen students’ identity formation and clinical confidence, while supporting continuity between generations.

However, participants expressed concern that practical midwifery skills are at risk of being lost. The declining emphasis on normal birth, the marginalization of midwives in leadership roles, and the growing academicization of midwifery education contribute to a detachment from core practices. There is an emerging sense of uncertainty about what legacy will be passed to the next cohorts of midwives, as the unique knowledge and skills of the midwife “by the woman’s side” is increasingly being replaced by the midwife “by the obstetrician’s side.” [31,32,33]. Given this shift, it is essential to identify the fundamental skills and competencies that are needed to effectively and competently perform the midwife’s role, underscoring the need for skill development and assessment processes [14,34].

Consistent with the previous literature, participants viewed the biomedical model’s dominance and the over-medicalization of childbirth as major threats to midwifery’s integrity and autonomy [12,23]. Increasing reliance on technology and risk-based approaches has led to unnecessary interventions, reinforcing misconceptions about midwives’ competencies and undermining woman-centered care [35,36,37]. These findings are consistent with studies from other countries, such as Australia, Italy, Sweden, and other European contexts, where midwives similarly report tension between biomedical dominance and midwifery-led care models [38,39,40]. International evidence highlights that midwives’ visibility, autonomy, and professional satisfaction increase when supported by policies that promote continuity of care and community-based practice [41].

Despite these challenges, midwives reported a strong sense of professional identity rooted in their cultural legacy and collective memory. Being “with woman” was described not just as a practice, but as an integral part of their identity. This relational approach to care highlights a deeper commitment to the mother–newborn dyad and distinguishes midwifery from other health professions [42].

The profession, however, continues to face devaluation by both society and institutional systems, in comparison to the medical profession [39], and this often results in low levels of job satisfaction [43]. Cypriot midwives have identified organizational barriers such as poor managerial support and understaffing as detrimental to their effectiveness and morale [14]. Conversely, when midwives feel organizational support and have adequate staffing, they report greater satisfaction and are more likely to remain in the profession [34,44,45]. Midwifery leadership was also noted as a key factor to foster empowerment by promoting autonomy, professional visibility, and the transfer of experiential knowledge. Through mentoring and role modeling, leaders enable skill exchange between generations, strengthen professional identity, and support midwives’ full scope of practice [24,46,47,48].

Participants stressed that for midwifery to thrive and for women to be truly empowered, national health policies must acknowledge midwives’ full scope of practice [49]. This includes community-based maternity care, continuity of care models, and comprehensive antenatal education. These models have been associated with stronger professional identity, increased autonomy, job satisfaction [8,46], better outcomes for mothers and babies [14,50], and reduced attrition from the profession [51] and support the development of competencies among midwives in training [30]. Ensuring the availability of sufficient and well-trained midwives should be part of the national health policy and planning [52]. Lastly, cultural competence must be recognized as a core component of midwifery education and practice in Cyprus, especially given the island’s growing social and cultural diversity [53]. National midwifery bodies, such as the Midwives Committee of the Cyprus Nurses and Midwives Association, have a vital advocacy role in promoting midwives’ autonomous practice as defined by law and in ensuring their visibility and authority within the healthcare system. To fully realize their potential, midwives must also be actively involved in health policy formulation and decision-making, ensuring that their voice shapes the future of maternity care [48,49].

This study offers new insights into how midwives and student midwives perceive the intangible heritage of midwifery and how historical and institutional contexts shape midwives’ identity. While midwifery identity has been previously explored in relation to mentorship, clinical models, and professional development, this study frames the issue through an additional conceptual lens. To our knowledge, it is the first study to explore how midwives and student midwives in Cyprus perceive midwifery’s intangible heritage, offering a unique lens to understand the intergenerational transmission of values and skills, particularly within a medicalized maternity system.

### 4.1. Ethics

The study was conducted in accordance with the ethical principles outlined in the Declaration of Helsinki. Ethical approval was obtained from the Cyprus National Bioethics Committee (Reference No. 2023.01.84) and the Departmental Postgraduate Committee of the Nursing Department at the Cyprus University of Technology. Participation was entirely voluntary, and all participants were provided with detailed information about the study before signing a written informed consent form. Consent included permission to record, transcribe, and analyze the discussions, with assurances of confidentiality and anonymity throughout the research process. Personal identifiable information was omitted from the transcripts, and only the lead researcher had full access to the data.

### 4.2. Limitations

While the study offers rich insights, the findings represent the perceptions of midwives and student midwives from a single country and are thus not generalizable. However, the findings offer valuable suggestions of urgent action needed to preserve the role of midwifery, or in fact reintroduce it in its full scope, in settings of an overmedicalized birth environment. Furthermore, while data saturation was reached with the three focus groups, theoretical saturation may not have been fully achieved due to the purposive sampling approach. Although participants varied in terms of years of professional practice, they were mostly from the same district, and most were employed in the public sector. Furthermore, they were familiar with each other, due to the small size of the country and the small number of midwives who often meet at national conferences and seminars. Student midwives were enrolled at the same university; hence, they had the same influences from their academic tutors and clinical mentors. Overall, it is unclear whether efforts to recruit a more heterogenous sample would provide additional insights, especially if more midwives from the private sector participated.

## 5. Conclusions

Midwives in Cyprus feel that they have a heritage, a transferred legacy from one generation of midwives to another, and this makes them feel that they have a sense of common professional identity. However, systemic barriers such as medicalization, weakened leadership, and institutional undervaluation continue to threaten the preservation of MIH. Maternity care is a fundamental element not only for the existence of midwives but also for the health and well-being of women, babies, families, and, ultimately, society as a whole. More than an academic discipline, midwifery involves values, attitudes, skills, and practices that define the profession. To preserve this heritage, it is important to nurture the bond between generations of midwives in order to maintain the legacy and sense of common professional identity and pass expertise from one generation to the next.

Immediate action is needed to strengthen, protect, and advance the midwifery profession. Strategic priorities should include promoting role modeling within the profession, fostering effective and empowering leadership, advocacy for the physiology of birth, and empowering women to seek midwifery care within an increasingly medicalized birth environment.

## Figures and Tables

**Figure 1 healthcare-13-01936-f001:**
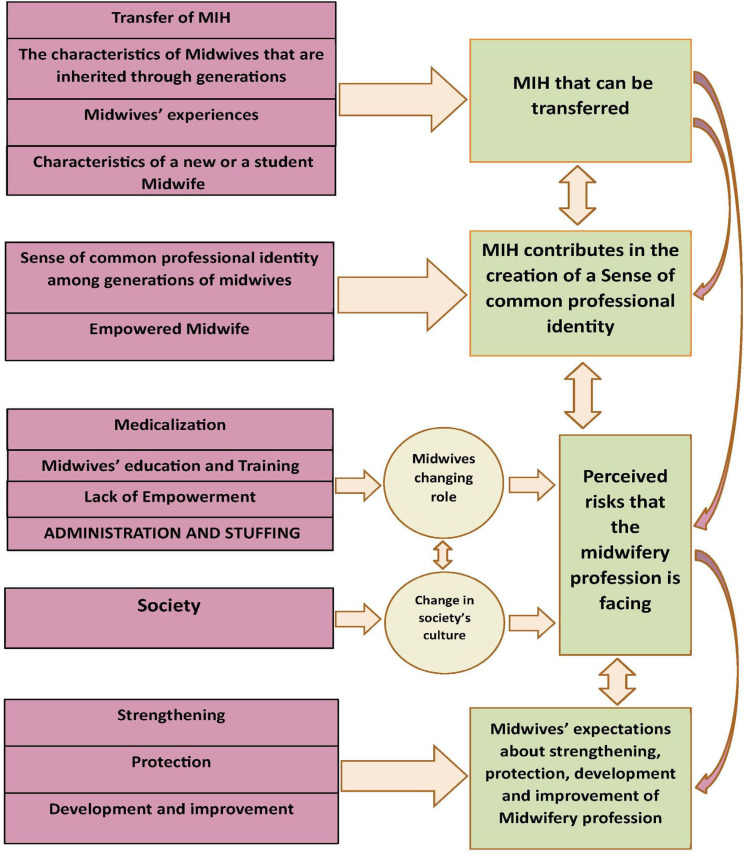
Thematic Analytic Schema to draw the links between categories and themes.

**Table 1 healthcare-13-01936-t001:** The agreed final version of the guide, consisting of 10 questions grouped into three thematic areas.

	List of Possible Questions (High Importance Questions, Questions that Provoke the Discussion)
Experiences	1. Talk to us about your experience as a midwife/student midwife about the way that MIH is transferred
	2. What was the element that helped you the most during the procedure of understanding midwifery practice and science?
	3. Beyond university, what are your experiences in the clinical field and your contact with more “older” experienced midwives? a. If you wish, you can share with us specific incidents, events, experiences, anything that impressed you, etc. b. Do you believe that there is MIH that is passed through generations of midwives?
Perceptions	1. What gives midwives a “sense of professional identity”? Can you describe what it is that connects midwives–that creates a sense of common professional identity?
	2. What do you believe is the Cypriot Society’s view of the midwifery profession? 3. What do you understand by the concept of MIH? Do you think this is linked to the midwife′s sense of professional identity? 4. What are the elements that prepare a midwifery student/new midwife to practice the profession?
Expectations	1. Can you tell us in what ways the midwifery profession in Cyprus could be protected and developed (risks)?
	2. What could be the recommendations for improvement changes so that the profession of midwifery emerges again as an integral, elemental foundation of better perinatal care and a prosperous society?
	3. Anything else you would like to add, which we did not cover above? Or that you have the chance to say?

**Table 2 healthcare-13-01936-t002:** Steps and actions taken for analyzing, coding, and categorizing codes into themes.

Steps	Actions
1	Development of an introductory section for understanding the theme
2	Recording and transcribing
3	Dada coding
4	Making links with the literature
5	Grouping of similar codes and supporting interview extracts into categories
6	Codes and categories analysis: revision of codes, comparing differences and similarities, exchanging ideas and interpretations
7	Synthesis of categories into themes
8	Final agreement on the themes in relation to the literature and the available evidence
9	Selection of quotes illustrating the data analysis and the synthesis of the themes

**Table 3 healthcare-13-01936-t003:** Status and years of experience of the participants.

Age	Status	Years of Experience
20–30	7 Nurses/student midwives	1 year of experience in practicing midwifery as SM
30–40	5 Midwives	6–10
40–50	3 Midwives	10–15
50–60	3 Midwives	15–20
Over 60	4 Midwives	Over 30

## Data Availability

Interview transcripts are confidential and cannot be shared publicly in order to protect participant anonymity, in line with the approved ethics protocol. De-identified data excerpts are available from the corresponding author upon reasonable request, subject to ethical approval.

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
