# Peer review of "“What You Leave…Will Leave You”: A Qualitative Study of Perceptions of Midwifery’s Intangible Heritage and Professional Identity Among Midwives and Student Midwives in Cyprus"

_healthcare, 2025, doi:10.3390/healthcare13151936_

Round 1

Reviewer 1 Report

Comments and Suggestions for Authors

Please find below a structured summary of the reviewer’s comments to assist in the revision of the manuscript. The comments are categorized by manuscript sections for clarity and highlight specific areas for improvement, missing information, and required formatting adjustments.

Author Response

Dear reviewer,

Please find attached the revised manuscript along with a detailed response to the reviewers' comments. We have addressed all suggestions to the best of our ability and have highlighted the changes accordingly. We truly appreciate the constructive feedback, which helped us improve the clarity and quality of our work.

Thank you for your time and consideration.

Sincerely,
Maria Panagiotou
On behalf of all co-authors

Reviewer 2 Report

Comments and Suggestions for Authors

As attached 

Author Response

Dear Reviewer,

Please find attached the revised manuscript along with a detailed response to the reviewers' comments. We have addressed all suggestions to the best of our ability and have highlighted the changes accordingly. We truly appreciate the constructive feedback, which helped us improve the clarity and quality of our work.

Thank you for your time and consideration.

Sincerely,
Maria Panagiotou
On behalf of all co-authors

Reviewer 3 Report

Comments and Suggestions for Authors

Dear Authors,

Theme of your manuscript is interesting, as it is worth to depicture the perspective of midwifes from the country with historical background.

I would like you to read and take an attitude / clear my doubts.

  1. One of the theme is excessive medicalization, which is presented along with the doctors focus on performing CS, and attributing the obstetricians with faults of lack of postnatal care (362) - are there any legislative issues concerning postnatal care or is based on patients' decisions? Some sentences concerning the subject of cooperation in manuscript might be perceived conflictive. Try to avoid them. 
  2. Can you point if there are legislations for low-risk antenatal care in your other countries (375)? Is it restricted to antenatal classes or providing prenatal routine pregnancy monitoring? You can refer to WHO Midwife-Led Continuity of Care or other healthcare recommendations. Providing informations about prenatal care in Cyprus might be subject of interest to midwifes from different countries. 
  3. How educational schema to be a midwife in Cyprus looks like? Is it 3 / 5 years of University or training collage? 
  4. Overmedicalization and doctors' interventions is perceived as a great threat, but maybe it is worth to show rational medicalization and cooperation (with recommended by healthcare recommendations CS percentage) as a positive factor or lowering the risk of mother and child morbidity in cases of high-risk pregnancies and deliveries. Emphasizing positive aspects with reference support might improve overall sound of the manuscript.
  5. Abbreviations:  add CS.
  6. Change an order of references - line 409.
  7. References: please use same alphabet for Greek articles. 

Author Response

(The authors gave the same response as above.)

Round 2

Reviewer 1 Report

Comments and Suggestions for Authors

Dear Authors,

Thank you for your efforts in revising the manuscript. I appreciate your careful attention to the feedback provided. The changes you have made have significantly improved the clarity and quality of the paper. I have no further suggestions at this point. I wish you success with the publication process.

Warm regards,